# Impaired Human Sexual and Erectile Function Affecting Semen Quality, in Obstructive Sleep Apnea: A Pilot Study

**DOI:** 10.3390/jpm12060980

**Published:** 2022-06-16

**Authors:** Konstantina Kyrkou, Emmanouil Alevrakis, Katerina Baou, Manos Alchanatis, Cornelia Poulopoulou, Christos Kanopoulos, Emmanouil Vagiakis, Dimitris Dikeos

**Affiliations:** 1Department of Psychiatry, General Panarkadian Hospital of Tripoli “Evangelistria”, 22100 Tripoli, Greece; kelinagr@gmail.com; 2Fourth Department of Respiratory Medicine, Sotiria Hospital of Chest Diseases, Mesogeion 152, 11527 Athens, Greece; katebaou@gmail.com; 3First Department of Respiratory Medicine, National and Kapodistrian University of Athens Medical School, 11527 Athens, Greece; nalxanat@gmail.com; 4First Department of Neurology, National and Kapodistrian University of Athens Medical School, Eginition Hospital, 11528 Athens, Greece; cpoulopoul@gmail.com; 5Biopathological Diagnostic Research Laboratory “Analysi”, Vas. Sofias 74 St., 11528 Athens, Greece; ckanopoulos@gmail.com; 6Critical Care and Pulmonary Services, Sleep Disorders Center, National and Kapodistrian University of Athens Medical School, Evangelismos Hospital, 10676 Athens, Greece; emvagiakis12@gmail.com; 7First Department of Psychiatry, National and Kapodistrian University of Athens Medical School, Eginition Hospital, 11528 Athens, Greece; ddikeos@med.uoa.gr

**Keywords:** obstructive sleep apnea, sleep disorders, fertility, male infertility, erectile dysfunction, polysomnography

## Abstract

Obstructive sleep apnea (OSA) is a rising problem, with important implications for public health. Recent evidence has revealed a link between OSA and reduced male fertility. We investigated the association between OSA and sexual and erectile function, as well as semen quality, and the effect of treatment by continuous positive airway pressure (CPAP). A total of 41 male subjects, who underwent polysomnography for suspected OSA, participated in the study. Erectile and sexual function were assessed with the 15-item International Index of Erectile Function (IIEF-15) questionnaire, blood samples, and sperm analysis. OSA patients after the initiation of CPAP treatment were followed for a period of 1 year. Thirty-two patients were diagnosed with OSA, and nine subjects without OSA were used as a control group. OSA patients demonstrated significantly impaired erectile function, reduced testosterone levels, and lower semen quality. Multivariable regression analysis showed that BMI and IIEF score were independent determinants of AHI. Sexual function improved after a year of CPAP therapy in OSA patients. This study provides further evidence regarding the association between OSA and erectile function impairment, as well as semen quality. Longitudinal adherence to CPAP treatment has a beneficial effect on erectile function.

## 1. Introduction

Male infertility is a common problem with widespread societal consequences, with the percentage of infertile males ranging from 2.5 to 12% in different studies and different parts of the world [1]. The observation that sperm counts might be gradually declining worldwide is even more worrisome [2], even though the methodology of the study that came to this conclusion has several weaknesses [3]. The association between Obstructive Sleep Apnea (OSA) and male infertility has started to come to light in recent years, with a significant number of studies both in animals and humans demonstrating a link between OSA, intermittent hypoxia, and sperm measures [4].

Many studies have concluded that there is a clear negative association between sleep disturbances and male fertility, although which exact aspects of the latter are affected is less clear. There is a near ubiquitous agreement in the literature that sleep disturbances, including documented obstructive sleep apnea, bad self-reported sleep quality, shorter overall sleep duration, and even later bedtimes are associated with disturbances in semen quality and quantity, with OSA patients presenting reduced sperm volumes and/or concentrations and fewer normal spermatozoa, with reduced motility and vitality [5,6,7,8,9,10]. Another common association is that between sleep disturbances and impaired erectile and overall sexual function. In a study by Budweiser et al., 69% of males diagnosed with OSA also self-reported erectile dysfunction (ED), with lower mean nocturnal SaO2 being an independent risk factor, and overall higher sexual dysfunction scores [11]; in addition, a recent meta-analysis found a lower risk of ED in men without OSA [12]. The effect of sleep disturbances on male reproductive hormones is less clear, with some studies demonstrating a link between OSA and reduced testosterone but no other reproductive hormone [13].

The association between the two entities becomes even clearer when considering the effect that OSA treatment, namely CPAP, has on male fertility. Treatment with CPAP has been shown to improve erectile function [13,14,15], with some authors even reporting a 75% ED remission rate at one month [16]. Additionally, some studies have reported an improvement in the male hormone profile after CPAP treatment [14,17], while others have found no such association [13,18]. Interestingly, there is a paucity of studies regarding the effect of CPAP treatment on sperm quantity and quality.

The principal aim of this study is to investigate possible associations between OSA and male sexual function and fertility by embracing a global approach that includes laboratory sleep studies, hormone levels, questionnaires, and sperm analysis.

## 2. Materials and Methods

This is a prospective study with a population of 41 male subjects. Any consecutive male patient, aged between 18 and 60 years, with suspected OSA was considered a potential candidate. The suspicion of OSA was based on symptoms such as morning tiredness, sleepiness, and/or excessive snoring. Subjects were referred to the Sleep Disorders Centers of Evangelismos and Eginition hospitals, over an 18-month period from January 2016 to January 2018, and the OSA diagnosis was established by polysomnography. A thorough history which included socio-demographic and sexual life information, medical history, lifestyle, occupation, and sleep habits was taken by a physician. Shift workers were excluded from the study. Fatigue Severity Scale (FSS) [19], Epworth Sleepiness Scale (ESS) [20], Hospital Anxiety and Depression Scale (HADS) [21], and Pittsburgh Sleep Quality Index (PSQI) [22] were completed by the participants. Subjects were asked to complete the 15-item International Index of Erectile Function (IIEF-15) questionnaire [23], and they were also asked to rate from 0 to 10 their sexual desire, their penile hardness, control of ejaculation, and the duration of sexual intercourse. Patients who were diagnosed with OSA and accepted to use a CPAP apparatus were followed and re-examined at 3, 6, and 12 months after the baseline examination. The study complies with the Declaration of Helsinki. The study protocol was approved by our Institutional Research Ethics Committee and all subjects gave informed consent.

### 2.1. Polysomnography

Participants underwent a split-night study to diagnose OSA and titrate CPAP. Polysomnography was performed using standard techniques, as have been previously described by the American Academy of Sleep Medicine [24]. Recorded channels included electroencephalograms, bilateral electrooculograms, electrocardiogram, chin electromyogram, thoracic and abdominal respiratory inductance plethysmography, airflow (via oral/nasal thermistor and nasal pressure transducer), oxyhemoglobin saturation (finger pulse oximetry), leg movements, and body position. Studies were scored by two experienced sleep technicians and secondarily reviewed by a sleep physician following standard criteria.

Events were classified as obstructive based on the presence of respiratory effort as measured by thoracic and abdominal plethysmography. An apnea was defined as a complete cessation of airflow for more than 10 s. A hypopnea was defined as a ≥30% reduction in airflow or thoracoabdominal movement for 10 s and accompanied by ≥3% oxygen desaturation and/or an arousal. Oxygen desaturation events were automatically scored from the oxygen saturation channel and manually edited for artifact. CPAP titration was performed, manually, using a uniform approach. Titration began usually at a pressure of 4 cm H_2_O. The pressure was increased gradually every five or more minutes in order to eliminate obstructive apneas, hypopneas, and eventually snoring. However, if, with increasing pressure, central apneas appeared, the pressure was reset to the previous level, where central apneas were not present.

### 2.2. Laboratory Tests

Samples were collected after an overnight fast, before the initiation of CPAP therapy. Serum was separated after centrifugation and stored at −20 °C until analyzed. Testosterone levels were measured by chemiluminescence using the Siemens ADVIA Centaur Testosterone II (TSTII) assay with the ADVIA Centaur^®^ CP analyser (Siemens Healthcare Diagnostics Inc., Tarrytown, NY, USA).

After semen collection, semen analysis was obtained. The number of sperm per milliliter of ejaculation and sperm morphology and motility were measured. Analyses mainly included: (1) sperm morphology, which was viewed in fresh specimens and a 1:10 dilution, using a Zeiss Axiostar Plus microscope at a magnification of 400 and was expressed as a percentage of normal sperm having an oval head with a long tail; (2) sperm count, expressed as the number of sperm per milliliter (mL) of semen in 1 ejaculation; (3) sperm motility, which is sperm swimming forward progressively, expressed as a percentage of motile sperm according to 2010 WHO criteria [25].

### 2.3. Statistical Analysis

Data were analyzed using SPSS 15.0 (SPSS, Chicago, IL, USA). Normality was tested with the Kolmogorov–Smirnov criterion. Continuous variables are expressed as means ± S.D. Skewed variables are expressed as medians (interquartile range). Comparison of continuous parameters between the groups of patients and controls was carried out with the unpaired Student’s *t*-test or the nonparametric Mann–Whitney U test for normally distributed and skewed variables, respectively. The association of categorical clinical variables was assessed with the χ^2^-test. Correlations between continuous variables were evaluated with the Pearson’s or the Spearman’s correlation coefficient for parametric and non-parametric variables, respectively. Multivariable linear regression analysis was applied to evaluate the association between AHI as dependent variable and testosterone, and IIEF as independent variable, after adjustment for potential confounders (age and BMI). Repeated-measures analysis of variance was performed to evaluate the effect of PAP treatment. Values of *p* < 0.05 were considered statistically significant.

## 3. Results

### 3.1. Subject Characteristics

The study population consisted of 41 subjects: 32 patients were diagnosed with OSA, and 9 subjects without OSA were used as a control group. Clinical characteristics of the study groups are shown in Table 1. OSA patients and controls had similar age, and frequency of smoking, diabetes mellitus, arterial hypertension, and hypothyroidism. Furthermore, the presence of sleepiness, fatigue, anxiety, and depression, according to the questionnaires, did not differ between the two groups. OSA patients compared to subjects without OSA had higher BMI, and poorer sleep quality (Table 1).

### 3.2. Relationship between OSA with Erectile and Sexual Dysfunction

Erectile function, according to IIEF score, sexual desire, and the subjective feeling of penile hardness, control of ejaculation, and duration of intercourse were significantly impaired in OSA patients compared to control subjects. OSA patients had significantly reduced levels of testosterone compared to controls. Sperm concentration and total sperm count did not differ between the two groups, but OSA patients had significantly reduced percentage of motile sperm compared to control group (Table 2). In the whole study population, AHI was significantly correlated with testosterone levels (*r* = −0.379, *p* < 0.05) and IIEF score (*r* = −0.533, *p* <0.001). Multivariable regression analysis showed that BMI and IIEF score were independent determinants of AHI (Table 3).

### 3.3. OSA Patients and CPAP Usage

Among the 32 patients with OSA, 25 patients initiated the use of the CPAP apparatus. Characteristics of the two groups are shown in Table 4. OSA patients showed an improvement in their sexual functioning after the treatment with CPAP. The results of repeated measure analysis of variance are shown in Table 5.

## 4. Discussion

### 4.1. Erectile Function

The results of our study further corroborate the findings of previous reports about the relationship with male infertility and OSA. Parameters of sexual function were worse in the OSA group compared to the control group, a consistent finding in most studies [11,12,13,16]. At baseline, CPAP users reported lower sexual desire compared to non-CPAP users, while other parameters such as the IIEF total score, self-reported penile hardness, ejaculation control, and duration of intercourse did not differ significantly between the two groups. Patients who started using CPAP demonstrated significant improvements in the last four parameters over a period of 12 months, with the effect plateauing after 3–6 months.

### 4.2. Semen Analysis

Even though our study did demonstrate differences in sperm motility between the OSA and non-OSA patients, there were no differences in sperm volume and concentration. This is in contrast to a number of studies demonstrating an association between sleep disturbances and both quantitative and qualitative disturbances in sperm [5,6,7,9]. There are marked methodological differences between most studies. To begin with, most available studies associate sperm characteristics with sleep disturbances in general, usually assessed with questionnaires and very rarely reporting polysomnography results, rather than OSA per se. Even then, different questionnaires are used between studies such as the four-item version of the Karolinska Sleep Questionnaire [9], the PSQI [5,6], or other questionnaires [10]; some studies even report exclusively on parameters such as sleep duration and time in bed [7]. In our case, it should be noted that the control group of our study included patients who were referred to the sleep lab with the suspicion of OSA and most of them self-reported disturbances in sleep quality. Indeed, the control group of non-OSA patients demonstrated a median PSQI of 8 (IQR 7–8.5), and all subjects in the control sample (100%) had a PSQI score > 5, which is consistent with poor sleep quality, even though OSA patients had significantly higher PSQI scores. Nevertheless, there is a solid experimental foundation on murine models connecting sleep disturbances with male infertility and more specifically disturbances in sperm motility and viability [26,27,28,29], as well as ED [30].

### 4.3. Testosterone Levels

Patients with OSA had significantly lower levels of testosterone at baseline compared with non-OSA controls. This finding seems to add to the confusion in the current literature where studies do not consistently demonstrate differences in baseline testosterone levels or after CPAP treatment in this population [18]. This is also true of experimental murine models of sleep deprivation [30]. One possible confounding factor for this finding is the higher prevalence of obesity in the OSA group since both ED and lower testosterone levels are known to be more prevalent in obese individuals [31]. Despite that, the IIEF score was independent of BMI in our regression analysis.

### 4.4. Pathophysiology

There are many physiological pathways connecting OSA and sleep disturbances with ED and semen perturbations. Paradoxic sleep deprivation (PSD) in rats has been demonstrated to dampen sexual activity, reducing the number of intromission attempts, successful intromissions, and ejaculations. This is accompanied by significantly lower testosterone levels in the PSD compared to the control group, which could potentially explain the finding [26]. As mentioned previously, lower mean nocturnal SaO_2_ is an independent risk factor for ED in humans [11] and at least one study has demonstrated that CIH was associated with a reduction of endothelial nitric oxide synthase (eNOS) expression in rat erectile tissue [30]. Hypoxia is known to exert a profound negative effect on erectile function in humans, a finding that can be attributed to studies performed at higher altitudes where the partial pressures of inspired oxygen are lower. In those settings, sleep-related erections are characterized by significantly reduced erectile rigidity, presumably through the negative effect of hypoxia on NOS activity [32]. Other proposed mechanisms that primarily contribute to disruption of spermatogenesis include oxidative stress on the testes [29], damage to the endothelium, as well as disruption of the blood–epididymis barrier [28]. Another study has also suggested a possible implication of antisperm antibodies, with humans who sleep for less than 6 h presenting a higher positivity rate [7].

### 4.5. Limitations

This study is subject to certain limitations that need to be acknowledged; first and foremost, the sample size is small due to difficulties in recruiting patients for a demanding study requiring semen samples from patients with no known infertility. The original purpose of the study was to also document repeat semen samples after three months of CPAP implementation, but the number of available repeat samples was too small to be included in the study. Another issue of the study, as analyzed above, was the use of patients who reported sleep disturbances but were not found to suffer from OSA, as a control group, as well as the significantly higher BMI in the OSA group. It would also have been preferable to utilize an erectometer to objectively assess penile hardness, rather than use a self-assessment method. Notwithstanding that, it was deemed that utilizing an erectometer would possibly further complicate an already demanding recruitment and follow-up process, or increase the dropout rate, so it was eventually considered more prudent to utilize questionnaires instead.

## 5. Conclusions

This study provides further evidence regarding the association between OSA and disturbances in male erectile function and semen quality. Additionally, patients with OSA had significantly lower testosterone levels than non-OSA controls. More importantly, we demonstrate that treatment with CPAP restores erectile function. Further research is required on subjects such as the effect of CPAP treatment on sperm quality and quantity, testosterone levels, and couple fertility when the male partner is suffering from OSA and thought to be contributing to infertility.

## Figures and Tables

**Table 1 jpm-12-00980-t001:** Clinical characteristics of the two groups (OSA patients vs. control group without OSA). Mean ± SD, unless otherwise specified.

	OSA Patients (*N* = 32)	Without OSA (*N* = 9)	*p* Value
Age (years)	46.4 ± 10.7	42.2 ± 8.4	0.287
BMI (kg/m^2^)	32.1 ± 5.2	27.5 ± 1.8	0.012
AHI (apneas/h)	55.3 ± 28.3	2.8 ± 0.7	0.001
Smoking *N* (%)	11 (34.4)	2 (22.2)	0.489
Diabetes mellitus *N* (%)	2 (6.3)	0 (0)	0.442
Arterial hypertension *N* (%)	8 (25)	0 (0)	0.095
Hypothyroidism *N* (%)	2 (6.3)	0 (0)	0.442
ESS	11.2 ± 5.7	9.1 ± 2.8	0.296
FSS	35.2 ± 15.8	30.2 ± 3.2	0.359
PSQI *	10.5 (7.3–12.8)	8 (7–8.5)	0.024
HADS *	10.5 (5–15)	10.78 (8.5–10.5)	0.816

* Median, interquartile range. Abbreviations: OSA, Obstructive Sleep Apnea; BMI, Body Mass Index; ESS, Epworth Sleepiness Scale; FSS, Fatigue Severity Scale; PSQI, Pittsburgh Sleep Quality; HADS, Hospital Anxiety and Depression Scale.

**Table 2 jpm-12-00980-t002:** Erectile and sexual function of the two groups (OSA patients vs. control group without OSA). Median, interquartile range, unless otherwise specified.

	OSA Patients (*N* = 32)	Without OSA (*N* = 9)	*p* Value
IIEF			
Total score	25 (22–28)	30 (30–30)	0.001
Sexual desire	7 (5–8)	10 (10–10)	0.001
Penile hardness	8 (6–9)	10 (10–10)	0.001
Control of ejaculation	9 (7–9)	10 (10–10)	0.001
Duration of intercourse	8 (6–9)	10 (10–10)	0.001
Testosterone (ng/dL) *	352.3 ±169.8	524.4 ± 170.3	0.019
Sperm concentration (c/mL × 10^6^)	40 (8.8–104)	48 (16–128.8)	1.000
Total sperm count (c × 10^6^)	89 (24.5–193.5)	98 (56–144)	0.796
Motile sperm % *	30.9 ± 23.2	53.6 ± 11.1	0.020
Immotile sperm % *	47.1 ± 24.9	29.3 ± 10.9	0.081

* Mean ± SD, Abbreviations: OSA, Obstructive Sleep Apnea; IIEF, International Index of Erectile Function.

**Table 3 jpm-12-00980-t003:** Multiple linear regression model evaluating the association of AHI with covariates.

	Unstandardized Beta Coefficient	Standardized Beta Coefficient	*p* Value
Model 1 (dependent variable: AHI) adjusted R^2^ = 0.217			
Age	−0.165	−0.051	0.749
BMI	2.220	0.334	0.035
IIEF	−2.432	−0.345	0.035
Model 2 (dependent variable: AHI) adjusted R^2^ = 0.147			
Age	0.293	0.091	0.549
BMI	2.596	0.391	0.016
Testosterone	−0.033	−0.179	0.244

Abbreviations: AHI, Apnea Hypopnea Index; IIEF, International Index of Erectile Function.

**Table 4 jpm-12-00980-t004:** Clinical characteristics of the two groups of OSA patients, according to CPAP use. Mean ± SD, unless otherwise specified.

	CPAP Users (*N* = 25)	CPAP Non-Users (*N* = 7)	*p* Value
Age (years)	47.1 ± 10.8	44 ± 10.9	0.522
BMI (kg/m^2^)	31.8 ± 5.4	33.6 ± 4	0.348
AHI (apneas/h)	62.7 ± 27.1	28.6 ± 11.9	0.001
Smoking *	9 (36)	2 (28.6)	0.715
Diabetes mellitus *	2 (8)	0 (0)	0.440
Arterial hypertension *	6 (24)	2 (28.5)	0.805
Hypothyroidism *	1 (4)	1 (14.2)	0.320
ESS	11.6 ± 6.1	9.7 ± 3.9	0.338
FSS	38.2 ± 14.8	24.4 ± 15.4	0.063
PSQI **	10 (7–13)	11 (8–13)	0.713
HADS **	11 (5–15)	8 (7–20)	0.721
IIEF score **	24 (21–26)	28.5 (26.5–30)	0.007
Sexual desire **	6 (4–8)	9 (6.7–10)	0.017
Penile Hardness **	8 (6–9)	9.5 (7.5–10)	0.060
Control of ejaculation **	9 (7–9)	9 (7.5–9)	0.364
Duration of intercourse **	7 (5.5–9)	9.5 (8–10)	0.067
Testosterone (ng/dL)	361.6 ± 178.9	319.2 ± 138.8	0.688
Sperm concentration (c/mL × 10^6^)	83.3 ± 77.6	25.4 ± 36.6	0.166
Total sperm count (c × 10^6^)	165.4 ± 183.3	52.9 ± 71.7	0.172
Motile sperm (%)	32.8 ± 22.3	22.5 ± 28.7	0.434
Immotile sperm (%)	44.4 ± 23.4	58.8 ± 31.9	0.300

* *N* (%), ** Median, interquartile range, Abbreviations: OSA, Obstructive Sleep Apnea; CPAP, Continuous Positive Airway Pressure; BMI, Body Mass Index; ESS, Epworth Sleepiness Scale; FSS, Fatigue Severity Scale; PSQI, Pittsburgh Sleep Quality; HADS, Hospital Anxiety and Depression Scale; IIEF, International Index of Erectile Function.

**Table 5 jpm-12-00980-t005:** Comparison of changes of sexual function (as assessed by IIEF) from baseline to follow-up in OSA patients with CPAP use. Repeated measures ANOVA; mean ± SD.

	Baseline	3 Months	6 Months	12 Months	*p* Value
Hardness	7.3 ± 1.9	8.8 ± 1.5	9.3 ± 1.2	9.2 ± 1.3	<0.001
Duration	6.9 ± 2.6	8.1 ± 2	9.1 ± 1.8	9 ± 1.8	<0.001
Ejaculation	7.8 ± 2.4	9 ± 1.3	9.2 ± 1.4	9.4 ± 1.2	<0.001
IIEF score	23.8 ± 3.8	27.4 ± 2.9	26.7 ± 4.1	27.9 ± 3	<0.001

Abbreviations: OSA, Obstructive Sleep Apnea; CPAP, Continuous Positive Airway Pressure; ANOVA, analysis of variance; IIEF, International Index of Erectile Function.

## Data Availability

The data presented in this study are available within the article.

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
