# Peer review of "Impaired Human Sexual and Erectile Function Affecting Semen Quality, in Obstructive Sleep Apnea: A Pilot Study"

_jpm, 2022, doi:10.3390/jpm12060980_

Round 1
Reviewer 1 Report
This study aimed to assess sexual function and semen analysis in patients with obstructive sleep apnea (OSA) and compare those with patients with sleep difficulty but without OSA.
The manuscript is well written, easy to understand, and suitable references have been used.
Line 81: write out the full term for the “IIEF-15” questionnaire on first use in the main text.
Author Response
Thank you for your kind remark. The suggestion correction has been made in the updated version of the manuscript.
Reviewer 2 Report
Interesting study and well-written paper. The authors could include whether they noted an association with ED and other parameters and oxygen saturation.
Author Response
Thank you for your remarks. As this association was not an outcome of the study, the data regarding oxygen saturation during the sleep studies is not immediately available.
Reviewer 3 Report
A well written article. It would be interesting to see how CPAP therapy affects the level of testosterone, sperm concentration, total sperm count, and sperm motility if this data is available. Even in a few patients (since this is a pilot study), if there is a table / data on the above levels before and after CPAP therapy will be extremely valuable to the readers.
Author Response
Your feedback is kindly appreciated. We agree that this sort of data would be valuable to the readers, unfortunately, due to the design of this study, such data was not collected. Nevertheless, the investigation of such a relationship posits an interesting subject for future studies.